# Impact of Iron Oxide on Anaerobic Digestion of Frass in Biogas and Methanogenic Archaeal Communities’ Analysis

**DOI:** 10.3390/biology13070536

**Published:** 2024-07-17

**Authors:** Xiaoying Dong, Aoqi Dong, Juhao Liu, Kamran Qadir, Tianping Xu, Xiya Fan, Haiyan Liu, Fengyun Ji, Weiping Xu

**Affiliations:** 1Liaoning Key Laboratory of Chemical Additive Synthesis and Separation, Panjin Institute of Industrial Technology, Dalian University of Technology, Panjin 124221, China; kamran_qadir@dlut.edu.cn (K.Q.); jifengyun321@126.com (F.J.); 2School of Petrochemical Engineering, Shenyang University of Technology, Liaoyang 111003, China; 15163873380@163.com (A.D.); jhliu0430@163.com (J.L.); 15890439402@163.com (X.F.); 3Panjin Institute of Industrial Technology, Dalian University of Technology, Panjin 124221, China; 13160605931@163.com; 4College of Chemical Engineering, Shenyang University of Chemical Technology, Shenyang 110142, China; 5Liaoning Key Laboratory of Chemical Additive Synthesis and Separation, Yingkou Institute of Technology, Yingkou 115014, China; lhy4486@yku.edu.cn; 6School of Chemical Engineering, Ocean, and Life Sciences, Panjin Institute of Industrial Technology, Dalian University of Technology, Panjin Campus, Panjin 124221, China

**Keywords:** anaerobic digestion, frass, iron oxide nanoparticles, biogas, methane, methanogens

## Abstract

**Simple Summary:**

This study investigates the impact of iron oxide (Fe_3_O_4_) nanoparticles on the anaerobic digestion (AD) of frass, a by-product of black soldier fly (BSF) larvae bioconversion of swine manure, and its co-digestion with corn straw. The aim is to enhance biogas production, particularly during the inoculum-free start-up phase of the AD process. Results indicate that the incorporation of Fe_3_O_4_ nanoparticles, especially those of ~184 nm size, significantly boosts average biogas yields, with a notable increase in methane production compared to control groups. The methanogenic analysis reveals that species such as *Methanocorpusculum*, *Methanosarcina*, and *Methanomassiliicoccus* play crucial roles in the AD reactor, with the ~184 nm group demonstrating optimal microbial diversity, potentially accounting for its superior gas output. The findings contribute to the advancement of renewable energy generation, sustainable development, and environmental conservation by providing a scientific basis for improving the efficiency of biogas production from organic waste.

**Abstract:**

With the increasing prominence of the global energy problem, socioeconomic activities have been seriously affected. Biofuels, as a renewable source of energy, are of great significance in promoting sustainable development. In this study, batch anaerobic digestion (AD) of frass (swine manure after bioconversion by black soldier fly larvae) and co-digestion with corn straw after the addition of iron oxide (Fe_3_O_4_) nanoparticles is investigated, as well as the start-up period without inoculation. The biochemical methane potential of pure frass was obtained using blank 1 group and after the addition of various sizes of Fe_3_O_4_ nanoparticles for 30 days period, and similarly, the digestion of frass with straw (blank 2) and after the addition of various sizes of Fe_3_O_4_ nanoparticles for 61 days period. The results showed that the average gas production was 209.43 mL/gVS, 197.68 mL/gVS, 151.85 mL/gVS, and 238.15 mL/gVS for the blank, ~176 nm, ~164 nm, and ~184 nm, respectively. The average gas production of frass with straw (blank 2) was 261.64 mL/gVS, 259.62 mL/gVS, 241.51 mL/gVS, and 285.98 mL/gVS for blank 2, ~176 nm, ~164 nm, and ~184 nm, respectively. Meanwhile, the accumulated methane production of the ~184 nm group was 2312.98 mL and 10,952.96 mL, respectively, which significantly increased the biogas production compared to the other groups. The methanogenic results of the frass (30 days) indicated that *Methanocorpusculum*, *Methanosarcina*, and *Methanomassiliicoccus* are the important methanogenic species in the AD reactor, while the microbial diversity of the ~184 nm group was optimal, which may be the reason for the high gas production of ~184 nm.

## 1. Introduction

The escalating global energy demand, driven by population expansion, underscores the imperative to transition from finite and environmentally harmful fossil fuels to sustainable solutions. Biogas emerges as an eco-friendly and economically beneficial alternative, poised to alleviate pressing energy supply challenges [1,2]. In recent years, the comprehensive production capacity of China’s animal husbandry sector has exhibited sustained growth. This expansion has been pivotal in ensuring national food security, bolstering the rural economy, and enhancing income generation for farmers and herdsmen. The sector’s advancements underscore its critical role in the broader agricultural framework and its contribution to socioeconomic stability. At the same time, it has also produced a large amount of livestock manure. It is reported that the annual production of swine manure in China reaches around 900 million tons [3]. While swine manure presents a risk of environmental contamination through direct or indirect pathways, it simultaneously holds promise as a valuable resource. Its judicious management and application are instrumental in safeguarding environmental integrity and promoting public health [4]. Meanwhile, In the agricultural sector, the annual production of straw-type materials has reached approximately 900 million tons. Developing efficient utilization strategies for lignocellulosic materials is critical for advancing sustainable agricultural practices and optimizing resource use [5]. In summary, the implementation of a global biomass production pathway has the potential to generate approximately 3000 TWh of electricity annually by 2050. This transition could result in an annual reduction of approximately 1.3 billion tons of carbon dioxide (CO_2_) equivalents, significantly contributing to global efforts in mitigating climate change [6].

Within the domain of biodegradable waste management, anaerobic digestion (AD) stands out as a highly efficient technology for waste treatment. This process not only mitigates environmental pollution but also generates renewable energy, contributing to the reduction in greenhouse gas emissions [7]. The process itself comprises four stages: hydrolysis, fermentation or acidogenesis, acetogenesis, and methanogenesis [8]. Following the aforementioned four steps, methane (CH_4_) is generated by organic waste conversion, which is a valuable energy source. One of the promising routes for anaerobic digestion process enhancement is via the incorporation of additives [9]. Trace metals, including cobalt (Co), nickel (Ni), iron (Fe), zinc (Zn), and molybdenum (Mo), serve as essential nutrient sources. These metals facilitate the synthesis of critical enzymes and co-enzymes, thereby enhancing the activity of anaerobic microorganisms during the anaerobic digestion (AD) process [7,10,11]. Trace metal additions boost organic substrate breakdown while also enhancing biogas and methane (CH_4_) generation [12]. The modification of nanomaterials could act as a safeguard, potentially offering a robust and effective remedial strategy to facilitate system recovery following instances of overload [13]. According to the reported findings, the average cell-to-cell distance in the Fe_3_O_4_-modified anaerobic system was greater than that in the non-amended control [14]. The electron transfer potential was stimulated by iron oxides via enriching syntrophic species [15]. It is reported that by using (1.5%) Fe_3_O_4_/GAC as an additive, high propionic acid consumption of up to 98% was achieved [16]. Recent research has delved into the impact of electro-conductive nanoparticles (NPs) when integrated into anaerobic digestion (AD) systems, examining how these particles induce changes in microbial community dynamics and syntrophic metabolism while also addressing the associated limitations and critical considerations [17,18].

Black soldier fly (BSF) larvae utilization is an effective waste disposal approach, following which energy and nutrients can be recovered for valuable product generation [19]. BFS larvae possess a remarkable innate capability to transform organic waste into nutritious insect biomass. These larvae feed on a wide array of organic materials, such as kitchen scraps, animal manure, and agricultural by-products, allowing them to thrive on diverse substrates. Through their digestive process, they are able to adapt to various waste streams, effectively converting waste into valuable bioresources [20,21]. Equipped with a distinctive digestive system and an insatiable appetite, they are capable of efficiently processing over 50% of organic waste biweekly under optimal conditions. This process not only contributes to waste reduction but also facilitates the production of biomass enriched with proteins and fats [22,23]. In contrast to traditional waste management methods (such as landfilling or composting), they have a low environmental impact, substantially lowering greenhouse gas emissions and pathogens [24,25,26]. Furthermore, BSF larvae are loaded with proteins and fats, making them a beneficial insect biomass source, and can be exploited as animal feed, feed additives, or as feedstock for biodiesel production [27,28,29]. However, the implementation of this methodology encounters multiple challenges and constraints. Primarily, the BSF larvae exhibit a constrained capacity to digest lignocellulosic materials, necessitating a pretreatment process to enhance the solubility and bioavailability of these residues for more efficient conversion [30]. Secondly, the efficacy of BSF larvae in processing diverse waste streams and the subsequent characteristics of the resulting biomass remain insufficiently explored within the academic domain. There is a pressing need for comprehensive optimization and in-depth investigation to elucidate these aspects [19]. Furthermore, the large-scale application of BSF larvae needs to be comprehensively assessed for the environmental impact on local ecosystems, biodiversity, and nutrient cycling [31]. In summary, BSF larvae can have significant advantages in waste management and biomass production utilization; however, rational planning, regulation, and monitoring for sustainable implementation are required.

Earlier anaerobic studies have investigated different types of organic waste, but the reports of frass utilization in anaerobic digestion disposal are few and sporadic. To the best of our knowledge, this study represents the first evaluation of the biogas potential of frass without inoculation. Batch experiments were conducted under mesophilic conditions to evaluate the biogas potential of standalone frass and frass combined with corn straw. Additionally, the influence of iron oxide nanoparticles on the anaerobic digestion of both substrates was systematically investigated. For comparison, experiments without and with ~176 nm, ~164 nm, and ~184 nm mean size Fe_3_O_4_ NPs were carried out. Biogas production, methane and carbon dioxide concentration, and the forms and quantities of methanogens were measured to investigate the AD process performance in full detail.

## 2. Materials and Methods

### 2.1. Synthesis of Iron Oxide Nanoparticles

A total of ~17.75 g of iron (II) chloride tetrahydrate (FeC1_2_·4H_2_O, Liaoning Quan Rui Reagent Co., Ltd., Jinzhou, China) and ~31.95 g of iron (III) chloride hexahydrate (FeC1_3_·6H_2_O, Liaoning Quan Rui Reagent Co., Ltd., Jinzhou, China) were measured to prepare a 250 mL mixed solution within pollen supernatant. Then, the mixture was stirred continuously and heated to 50 °C, followed by the addition of ~0.2 g sodium acetate (CH_3_COONa) and ~1 g polyethylene glycol (HO(CH_2_CH_2_O)_n_H). Then, 5 mL of sodium hydroxide (NaOH) was added dropwise under continuous stirring conditions until the pH was 10–11. Afterward, ~2.5 g of sodium bicarbonate (NaHCO_3_) was added and stirred for ~0.5 h. Finally, it was aged at 90 °C for 0.5 h, washed with deionized water, and then dried. The size of synthesized magnetite (Fe_3_O_4_) nanoparticles was adjusted through the concentration of a mixed solution of iron ions and the speed of the precipitation agent droplet when adjusting the pH value.

### 2.2. Feedstock and Its Characteristics

The frass utilized in this experiment was sourced from a medium-scale swine farm located in Shandong Province, China. This frass comprised a mixture of swine manure residues post-bioconversion by Black Soldier Fly larvae and insect excrements. Upon collection, the frass was promptly stored at 4 °C to preserve its integrity for experimental use. The experiment commenced without the addition of inoculum. Detailed characteristics of the feedstock are provided in Appendix A.

### 2.3. Experimental Setup

A schematic representation of the anaerobic digestion (AD) apparatus employed in this study is depicted in Appendix A. The batch lab-scale experiments were conducted using 1000 mL ground flasks, each with a working volume of 350 mL. These experiments were maintained under mesophilic conditions (approximately 35 ± 2 °C) using a water bath. Each flask was uniformly supplied with an appropriate amount of substrates, ensuring consistent initial total solids content (TS = 16.5%). The first group, the anaerobic fermentation group, was designed by three parallel groups (blank, ~176 nm, ~164 nm, and ~184 nm). In each bottle was added ~40 g of fresh frass and 200 mL distilled water (TS = 14.35%). The additional dosage of the iron oxide for the first fermentation group was ~150 mg, and ~176 nm, ~164 nm, and ~184 nm were added into the fermentation substrate separately before the start of the digestion of three parallel mixtures. Meanwhile, another three additive-free mixtures were used as the control (labeled as blank 1). The fermentation phase lasted for 30 days, and the heating water bath temperature was set at 35 °C. The second group was designed as an anaerobic fermentation group of ~176 nm, ~164 nm, and ~184 nm by three parallel groups. To each bottle was added ~95.5 g fresh frass, ~30.9 g corn straw, and 500 mL distilled water (TS = 19.49%). The additional dosage of the iron oxide for the second group was 400 mg, and ~176 nm, ~164 nm, and ~184 nm were added to the fermentation substrate separately before the start of the digestion with three parallel mixtures. In addition, another three additive-free mixtures were set as the control (labeled as blank 2). The fermentation phase lasted for 61 days, and the heating water bath temperature was 35 °C. 

Each flask containing substrates was purged with nitrogen gas for approximately 10 min to eliminate residual oxygen and ensure anaerobic conditions. Throughout the AD period, all flasks were manually agitated 5 times daily, with each agitation lasting for 2 min. Triplicate tests were performed for each experimental run to ensure reproducibility and accuracy.

### 2.4. Methods of Analysis and Calculation

The daily biogas production from each digester was quantified using the water displacement method. Biogas samples were collected in air cylinders, and both biogas production and methane content were measured daily. The methane content was analyzed by gas chromatography (A60, Panna). The pH value was measured using a pH meter (PHS-3C, Leici, Shanghai, China). The total solids (TS) and volatile solids (VS) were determined using standard analytical methods (APHA, 2005) [32]. Total C and N contents were quantified using a total carbon analyzer (multi N/C 2100). 

The micromorphology of iron oxide was examined using scanning electron microscopy (SEM, Nova Nano SEM 450, FEI, Hillsboro, OR, USA). X-ray diffraction (XRD, Lab XRD-7000s, Shimadzu, Shanghai, China) analysis was performed to determine the crystal structure and purity of the iron oxide sample, with a 2θ = 5–80°. The specific surface area of the iron oxide was measured using an absorption analyzer (ASAP 2020, Micromeritics, Shanghai, China), following the Brunauer-Emmett-Teller (BET) multipoint method of N_2_ adsorption.

### 2.5. Microbial Community Analysis

The influence of iron oxide on the methanogenic archaeal community dynamics during the AD of frass waste was examined using the Illumina MiSeq PE300 at Sangon Biotech Co., Ltd., Shanghai, China. Total community genomic DNA was extracted employing the E.Z.N.A™ Mag-Bind Soil DNA Kit (Omega M5635-02, Shanghai, China), in accordance with the manufacturer’s instructions. DNA concentration was quantified using a Qubit 4.0 (Thermo, Waltham, MA, USA) to ensure the extraction of sufficient high-quality genomic DNA. The V3–V4 hypervariable regions of the bacterial 16S rRNA gene were targeted for analysis. PCR was initiated immediately post-extracted, with the 16S rRNA V3–V4 amplicons being amplified using 2×Hieff^®^ Robust PCR Master Mix (Yeasen, 10105ES03, Shanghai, China). Hieff NGS™ DNA Selection Beads (Yeasen, 10105ES03, Shanghai, China) were utilized to purify the amplicon products by removing free primers and primer dimers. The samples were then sent to Sangon BioTech (Shanghai) for library construction using a universal Illumina adaptor and index. Before sequencing, the DNA concentration of each PCR product was determined using a Qubit^®^ 4.0 Green double-stranded DNA assay, and it was quality controlled using a bioanalyzer (Agilent 2100, Frederick, CO, USA). The first circle primers set of V3-V4 340FCCCTAYGGGGYGCASCAG, 1000R GGCCATGCACYWCYTCTC and the second circle primers set of 349F GYGCASCAGKCGMGAAW, 806R GGACTACVSGGGTATCTAAT were used for archaeal analysis. The Polymerase Chain Reaction (PCR) products from the initial PCR step were purified using VAHTS™ DNA Clean Beads. Subsequently, all PCR products were quantified using Quant-iT™ dsDNA HS Reagent and pooled together. Microorganism classification was performed using operational taxonomic units (OTUs) with a 97% identity threshold.

### 2.6. Statistical Analysis

To analyze the statistical differences among the experimental data, a one-way analysis of variance (ANOVA) was conducted using SPSS software (version 17.0). Statistical significance was determined at a threshold of *p* < 0.05.

## 3. Results and Discussion

### 3.1. The Characterization of the Iron Oxide Nanoparticles

In Figure 1a–c, the SEM images showed that the Fe_3_O_4_ has a regular morphology of predominantly cubic and some spherical nanoparticles with uniform size distribution. Figure 1d–f show the corresponding particle size distributions derived from the images. It shows that the mean sizes of nanoparticles are ~176.1 nm (a, d), 164.1 nm (b, e), and 184.1 nm (c, f). Although the pollen supernatant did not alter the morphology of the nanoparticles obviously, there was a significant increase (22%~28% improved compared to the blank) in the specific surface area of the ~176 nm, ~164 nm, and ~184 nm groups, which was 10.05 m^2^/g, 8.725 m^2^/g, and 7.69 m^2^/g, respectively. The XRD pattern of iron oxide is shown in Figure 1g. The diffraction peaks at [2θ] about 18.28°, 30.10°, 35.45°, 37.14°, 43.12°, 53.48°, 57.01°, 62.58°, 70.97°, 74.06°, 75.09°, 78.98°, and 86.77° are the characteristic peaks of Fe_3_O_4_ (JCPDS card No. 11-0614), which are attributed to (111), (220), (311), (222), (400), (422), (511), (440), (620), (533), (622), (444), and (642) planes, respectively.

### 3.2. Influence of the Iron Oxide on Gas Production of Frass 

As can be seen in Figure 2, the addition of magnetite of ~184 nm effectively increased the gas production of the frass from swine manure compared to the blank 1 group. From the 8th and 9th days, the gas production of each fermentation group began to increase significantly, and the daily gas production of the ~184 nm group was higher than that of the blank 1 group within 30 days. The addition of Fe_3_O_4_ in the form of nanoparticles reduced the lag phase, biostimulated the methanogenic bacteria, and increased their activity [33]. However, the ~176 nm and ~164 nm groups show inhibition on the frass anaerobic fermentation. Especially, the daily gas production of the ~164 nm group was lower than that of the blank group within 30 days (40.1 mL on average). The daily gas production of the ~176 nm group was slightly lower (8.2 mL on average) than the blank. The maximum daily biogas production is 288 L, which occurs on the 22nd day of the ~184 nm group.

The cumulative gas production of blank 1, ~176 nm, ~164 nm, and ~184 nm were 4383 mL, 4137 mL, 3178 mL, and 4984 mL, respectively. The average gas production rate of the blank 1 group and the nano group with ~176 nm, ~164 nm, and ~184 nm nanomaterials reached 209.43 ± 20 mL/gVS, 197.68 ± 17 mL/gVS, 151.85 ± 23 mL/gVS, and 238.15 ± 21 mL/gVS, respectively. Among these, the optimal effect of the fermentation was the group containing ~184 nm, which could increase the biogas production by 13.71% compared with the blank 1 group. In order to know the types and abundance of microbial species during the peak period, we opened the jar and took the fermentation samples for sequencing. Although the gas production stage was not completely finished, we calculated the gas production to be 30 days. The integration of Fe_3_O_4_ nanoparticles has been demonstrated to augment biogas yield while concurrently mitigating the accumulation of excess sludge in anaerobic digestion systems [34]. nZVI and Fe_3_O_4_ NPs boosted the hydrolysis-acidification process of the sludge [35]. In our experiment, the bigger the size of the nanoparticles, the better the effect of biogas production. Gabriella Papa et al. [36] investigated BSF reared on organic municipal solid waste, then anaerobic treatment of the remaining frass was performed, yielding biogas production of 138 NL (biogas)/g dry matter organic solid waste. Additionally, an analysis of the fluorescence spectra indicates that following the digestion by BFSL, there is a notable decomposition and transformation of the simple structural organic compounds (5.99~29.50%) present in animal manures into substances resembling humic materials [37]. It has been reported that the application of Fe_3_O_4_ nanoparticles at a concentration of 160 mg/L resulted in a biodegradability rate of 97.3%, compared to 51.4% observed in the control incubation [38].

As can be seen from Figure 2, the addition of different sizes of Fe_3_O_4_ effectively enhanced the methane content in the swine manure feces. From 10 d to 23 d, the methane content of the nano-added fermentation group began to rise sharply compared to the blank 1, and all nano-groups showed an increase in the methane content by 4.10%~19.73% compared with the blank 1 group. In 11 d~15 d, the methane content of ~176 nm (48.55%~58.14%) was significantly higher than that of the blank 1 group (24.45%~48.93%), which was also better than the ~164 nm group (47.27%~57.52%) and the ~184 nm group (44.18%~57.39%). From 16 d to 30 d, the methane content of the blank 1 group varied from 48.68% to 61.12%. In 16~25 d, the methane content of the ~176 nm addition group was the highest of all groups, which was among 58.46%~67.16%, and the highest methane content occurred on 21 d. From 18 d to 25 d, the methane content of ~164 nm was above 60%, and the highest methane content occurred on 22 d at 67.91%. From 16 d to 20 d, the methane content of the ~184 nm group was around 60%, and the highest methane content occurred on 30 d at 63.67%. Accumulated methane production of blank group 1 was 1881.18 mL within 30 d; meanwhile, the ~176 nm group was 1954.5 mL, the ~164 nm group was 1420.28 mL, and the ~184 nm group was 2312.98 mL, respectively. It was reported that smaller nanoparticles (~12–18 nm) usage led to a higher methane production boost than larger nanoparticles (~50–100 nm) [39]. Similarly, the addition of reasonably sized Fe2O3 nanoparticles helps improve the cumulative methane generation with reduced antibiotic resistance genes during AD of swine manure [40]. However, interestingly, in our experimental observations, the smaller nanoparticles (~164 nm) had an inhibitory effect on the methane production of frass. Harald Wedwitschka et al. [41] tested methane production utilizing the frass (obtained from swine manure by black soldier fly larvae (BSFL) feed from six kinds of crop feedstocks for anaerobic fermentation), and the results of biochemical methane potential (BMP) tests were 201 to 287 ± 37 mL (specific methane yield)/gVS, while semi-continuous pilot plant was 167 ± 15 mL (specific methane yield)/gVS.

From 2 d to 8 d of fermentation, the CO_2_ content of the ~176 nm group (5.46%~26.97%) and the ~164 nm nanomaterial addition group (5.14%~21.49%) was lower than that of the blank 1 group (11.47%~28.27%), which indicated that iron oxide had a certain inhibitory effect on the CO_2_ production of the fermentation in this period. The ~176 nm group had a promoting effect (38.86%~46.48%) in 9~15 d and had an inhibitory effect in 16 d~25 d, then a steering promotion effect in the last five days (26 d~30 d). The ~164 nm group presented a promoting trend in 10 d~15 d (improved 2.24%~25.31% compared to the blank 1), then presented an inhibitory trend in 2nd~9th (decreased 28.42%~57.06%) and second half of period (decreased 3.84%~32.34%). The ~184 nm group has an inhibitory effect (decreased 4.39%~20.41%) in the period except for several days (slight acceleration). Accumulated carbon dioxide production of blank 1 group was 1750.51 mL within 30 d; meanwhile, the ~176 nm group was 1551.43 mL, the ~164 nm group was 1131.51 mL, and the ~184 nm group was 1849.87 mL, respectively. Furthermore, ~164 nm has the strongest inhibitory effect on carbon dioxide production, which decreased by 35.36% compared to the blank. The greenhouse gas emissions were 0.0366 and 0.0290 kg CO_2_ eq./MJ elect for the control and Fe_3_O_4_ NPs, respectively [42], which is consistent with the results of this experimental study, the Fe_3_O_4_ had an inhibitory effect on CO_2_ production, it matched the international background of carbon neutrality and carbon peak.

### 3.3. Influence of the Iron Oxide on Gas Production of Frass with Corn Straw

The fermentation of frass with straw was carried out for 61 d. From 9 d to 20 d, the daily gas production of nanoparticle-containing groups was obviously better than blank 2, in which the ~176 nm group had the optimal cumulative gas production of 5717 mL, increased by 48.4% compared to blank 2 (2950 mL). From 21 d to 30 d, the cumulative gas production of the blank group was optimal (6512 mL). In the second half of fermentation, the daily gas production of the ~184 nm group was the best. And the maximum daily biogas production was 1034 mL, appearing on 26 d of the blank 2 group. As per reported results, Fe_3_O_4_ has promoted enzyme activities, and electron transfer in hydrolysis/acidification has enhanced the efficiency of hydrolysis and acidification [43,44]. We hypothesize that nanoparticles, characterized by their extensive surface area and a profusion of chemically active sites, may serve as an efficacious medium for the immobilization of enzymes. This immobilization is hypothesized to enhance both the stability and catalytic activity of the enzymes. Furthermore, nanoparticles are posited to play a pivotal role in the facilitation of electron transfer processes, which are integral to a multitude of metabolic reactions. Overall, the accumulated gas production of the blank 2 group was 18,315 mL, the ~176 nm nanomaterials group was 18,173 mL, the ~164 nm nanomaterials group was 16,906 mL, and the ~184 nm group had the highest gas production of 20,019 mL. The average biogas production rate of the blank group and nano-added groups (~176 nm, ~164 nm, and ~184 nm) was 261.64 mL/gVS, 259.62 mL/gVS, 241.51 mL/gVS, and 285.98 mL/gVS, respectively. Upon the completion of fermentation, the total solids (TS) content in the 176 nm group decreased by 15.99%, while the 164 nm group exhibited a reduction of 12.19%. Notably, the 184 nm group demonstrated the most effective TS removal, achieving a removal rate of 19.19%, which represents a 67.3% improvement compared to the control group.

As illustrated in Figure 3, the methane content in each nanoparticle-added group was significantly higher than that in the blank 2 group during the pre-fermentation period (prior to 21 d). Notably, on day 15, the methane content in the ~184 nm nanoparticle fermentation group reached 50%, which was 34.4% higher than the 15.6% observed in the blank 2 group. The methane content of the ~176 nm fermentation group was the highest among all the fermentation groups from 37 to 40 d and reached 72.71% on 38 d and 73.68% on 39 d, which were the highest values of methane content for all the fermentation groups. From 41 d onwards, the methane content of the ~176 nm fermentation group was between the ~164 nm and ~184 nm fermentation groups, higher than that of the ~184 nm fermentation group and lower than that of the ~164 nm fermentation group. It basically remained above 60% until 55 d. The methane content of the ~164 nm fermentation group was higher than the ~176 nm fermentation group and the blank within 18 days and lower than the ~184 nm fermentation group within 36 days. The promotion effect from 29 d to 36 d was not obvious, and even inhibition appeared at 33 d, 34 d, 38 d, and 39 d compared to the blank. However, the methane content of the ~164 nm fermentation group was ahead of the blank group and the other nanoparticle-added fermentation groups after 40 d and stayed above 65%. During the period of fermentation from 29 d to 36 d, the ~184 nm group had the highest methane content of all the gas-producing groups and reached the highest value of 70.99% on 35 d. The methane content of the ~184 nm group was more than 60% for in 29~45 d, and a mild inhibition was observed from 38 d to 40 d and from 51 d to 54 d. Accumulated methane production of the blank 2 group was 9237.62 mL within 61 d; meanwhile, the ~176 nm group was 9795.86 mL, the ~164 nm group was 9439 mL, and the ~184 nm group was 10,952.96 mL, respectively. The negatively charged Fe_3_O_4_ nanoparticles (NPs) exhibited the most pronounced positive effects on the AD of wheat straw, achieving a methane yield 51.33% higher than that of the blank control group, reaching 333.14 mL CH_4_/gTS [45]. Among the tested nFe_3_O_4_ concentrations (20, 50, and 75 mg/L), the 20 mg/L concentration demonstrated the highest hemicellulose degradation (93%) and methane yield (191.2 L/g VS) without posing any threat to anaerobic microorganisms [46].

On the first day of gas production in co-digestion of frass with straw, the daily CO_2_ content of the blank 2 group was 50.22%, and the daily CO_2_ content of each nano-added group was only 21.43%~32.43%, which was reduced by about 17.79%~28.79% compared to that of the blank 2 group. The CO_2_ content in each nanoparticle-added group was also significantly lower than that in the blank group during 22 d of fermentation, indicating that the addition of nanoparticles was beneficial in inhibiting the production of CO_2_. The daily CO_2_ contents of the blank and nano-added fermentation groups were similar from 23 d to 41 d, with contents ranging from 26.89% to 40.02%. In the fermentation period from 43 d to 61 d, the 20 nm fermentation group showed mild inhibition of CO_2_ compared to the blank 2 group, with the inhibition degree ranging from 0.88% in 42 d to 6.92% in 61 d. In the fermentation period from 49 d to 61 d, the ~164 nm fermentation group showed inhibition of CO_2_, with the degree of inhibition ranging from 0.52% to 17.39%. The degree of inhibition of CO_2_ at the end of fermentation was higher than that of the ~176 nm group by 10.47%. From 42 d to 61 d, ~184 nm nanoparticles promoted the CO_2_ production of the fermentation substrate, and the degree of promotion increased from 1.97% to 6.91% compared to the blank 2 group. Overall, the ~164 nm nanoparticles had a certain inhibitory effect on the production of CO_2_, which reduced the average CO_2_ content by 8.65%. The accumulated production of CO_2_ of blank 2 group was 6897.94 mL within 61 d; meanwhile, the ~176 nm group was 6546.65 mL, the ~164 nm group was 6300.93 mL, and the ~184 nm group was 7350.63 mL, respectively. 

### 3.4. Methanogen Activity Analysis 

As can be seen from Figure 4 and Table 1, the addition of Fe_3_O_4_ nanoparticles favored the diversity of methanogenic species. *Methanoculleus* is the absolute dominant species in each group. The addition of all three sizes of Fe_3_O_4_ nanoparticles effectively promoted the abundance of *Methanomassiliicoccus*. In particular, the addition of ~184 nm Fe_3_O_4_ increased the abundance of *Methanocorpusculum*, which was very poorly represented in blank 1. The abundance of *Methanosarcina* in the ~176 nm and ~184 nm sample groups was increased, which could explain the higher gas production rate in these two groups. *Methanocorpusculum* is favorable for the conversion of feedstock into biogas as the maximum gas production is 184 nm. It is reported that the quantity of *Methanosaeta* and *Methanosarcina* genera was observed to proliferate subsequent to the supplementation of Fe_3_O_4_ nanoparticles within the anaerobic digestion system designed for food waste treatment [15]. Upon the introduction of Fe_3_O_4_ nanoparticles into the anaerobic digestion process utilizing crude glycerol as a substrate, a significant upsurge in the abundance of *Methanomicrobiales* and *Methanosarcinales* orders was documented [18]. The nano-Fe_3_O_4_ biochar treatment was attributed to the enhancement of unclassified *Clostridiales* and *Methanosarcina* [47]. 

During anaerobic digestion, nanomagnetite may function as an electron acceptor, thereby accelerating the oxidation of volatile fatty acids and enhancing DIET as an alternative pathway for hydrogen/formate transfer between microorganisms. This mechanism can significantly augment methane production during anaerobic digestion [48,49]. As illustrated in Figure 4 and Table 1, the incorporation of conductive nanomaterials can modify the abundance of microbial populations, particularly key bacterial and archaeal groups involved in hydrolysis and methanogenesis. Specifically, the addition of nanomagnetite has been shown to increase the abundance of *Methanosaeta* and *Methanosarcina*, microorganisms that utilize electrons from the conductive material directly through the DIET process rather than via conventional hydrogenotrophic methanogenesis.

From Table 2, it can be seen that the Shannon values of the ~184 nm group are optimal, indicating the maximum gas production microbial diversity, which may be the reason for the highest gas production in the ~184 nm group. Secondly, the Shannon value of the ~176 nm group is 1.61; the third ~164 nm group is 1.60. Among them, the Shannon value of the blank 1 group is the lowest, which is 1.36, indicating the lowest microbial diversity. The OTUs clustering values of the blank group and the 100 nm group were higher, at 18 and 17, respectively. However, there is no positive correlation with biogas production. It was reported that 200 mg/L Fe_3_O_4_ NPs could promote the diversity of bacterial and archaeal communities [50].

From the sample correlation heatmap (see Figure 5), it can be seen that the correlation between the ~176 nm and ~184 nm groups and the ~164 nm and ~176 nm groups is 0.93. The correlation between ~176 nm and ~184 nm also reached 0.9. The nano group of ~184 nm had the highest correlation (value of 0.88) with the blank group. The correlation between ~176 nm, ~164 nm, and the blank 1 group was lower, with the values of 0.68 and 0.75, respectively. There are 17 common species between the blank 1 group and the ~184 nm group, 10 common species between the ~176 nm and ~164 nm groups, and 12 common species between the blank and the ~176 nm group. There are 10 common species between the ~164 nm and ~184 nm groups. There are 10 common species between ~176 nm and ~184 nm. The core microbial species mentioned above are *Methanoculleus*, *Methanomassiliicoccus*, and *Methanosarcina*. From FAPROTAX prediction, the hydrogenotrophic methanogenesis pathway is the main methane production type for all fermentation groups. Among them, the ~184 nm group had the highest abundance of this function, with 15,478 occurrences. It is reported that the integration of Fe_3_O_4_ nanoparticles within anaerobic digestion systems has a stimulatory effect on microbial communities, notably altering the ecological dominance toward hydrogenotrophic species [13]. Additionally, the biosynthesis of F420 was also promoted by it. In the Fe_3_O_4_ nanoparticles-based system, the CO_2_-reduced methanogenesis (as the main methanogenic pathway) was improved [51].

Secondly, the blank 1 group had 13,386 repetitions, while the ~176 nm and ~164 nm groups had 12,481 and 10,976 repetitions, respectively. In addition, the functional characteristics of the methane production pathway by reduction of methyl compounds with H_2_ were observed in all fermentation groups. But the addition of nanomaterials effectively enhanced the methane production pathway in this way, and this functional feature appeared 2623 times, 2110 times, and 2716 times in the predicted Table 3 at ~176 nm, ~164 nm, and ~184 nm, respectively, compared to the blank 1 group (325 times), which increased by 6.5–8 times. 

The prediction table also showed methanogenesis by disproportionation of methyl groups. The blank 1 group showed this functional feature 995 times, while the ~176 nm, ~164 nm, and ~184 nm groups showed 1436, 931, and 1521 times, respectively. From the data, the nanomaterials of the ~176 nm and ~184 nm groups also enhanced this methane generation pathway. An interesting phenomenon was found in the cumulative prediction table of the functional characteristics of these three main methane production pathways: the blank 1 group (14,381), ~176 nm group (13,917), ~164 nm group (11,907), and ~184 nm group (18,163). This value is positively correlated with the gas production of the corresponding fermentation group.

## 4. Conclusions

Swine manure after the bioconversion by black soldier fly larvae is an excellent fermentation raw material, and it can produce gas on its own without being inoculated as in this experiment. The results demonstrated that in both experiments, ~184 nm magnetic nanoparticles could improve the biogas production by 13.7% (frass) and 9% (frass with straw) compared to the control while increasing the methane production by 22.95% (frass) and 18.56% (frass with straw) compared to the control. The 176 nm increased the methane production by 3.89% (frass) and 6% (frass with straw) compared to the control but did not improve the biogas production and had an inhibitory effect on the production of CO_2_. The ~164 nm group showed inhibitory effects on the production of CH_4_ and CO_2_ in the pure frass group (frass) while slightly promoting CH_4_ production and inhibited CO_2_ production (frass with straw). The nanoparticles (164 nm and 176 nm) were beneficial for carbon neutrality, which could reduce carbon dioxide production. Finally, methanogenic archaeal communities’ analysis revealed that the optimal microbial diversity of methanogenic archaeal was the ~184 nm group. We observed that *Methanoculleus* and *Methanosarcina* had a positive correlation with biogas production. Utilizing insect frass for anaerobic fermentation to form a fully resourceful upstream and downstream process and achieve a maximum degradation of organic solid waste. 

## Figures and Tables

**Figure 1 biology-13-00536-f001:**
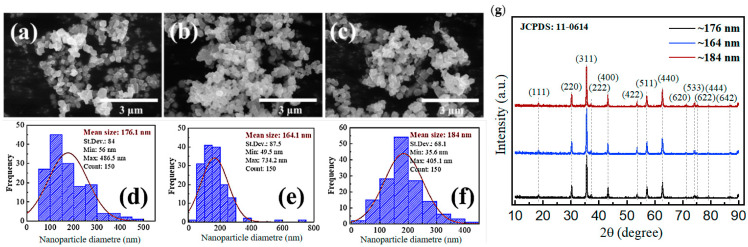
(**a**–**c**) The SEM images of Fe_3_O_4_. (**d**–**f**) Corresponding particle sizes distributions. (**g**) The XRD of Fe_3_O_4_ nanoparticles.

**Figure 2 biology-13-00536-f002:**
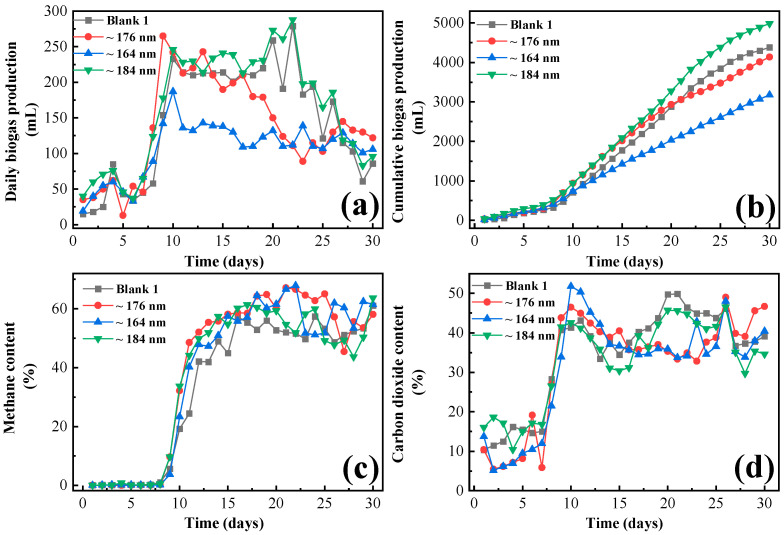
(**a**) Daily biogas production of insect frass; (**b**) Cumulative biogas production of insect frass; (**c**) Daily methane content of frass; (**d**) Daily carbon dioxide content of frass.

**Figure 3 biology-13-00536-f003:**
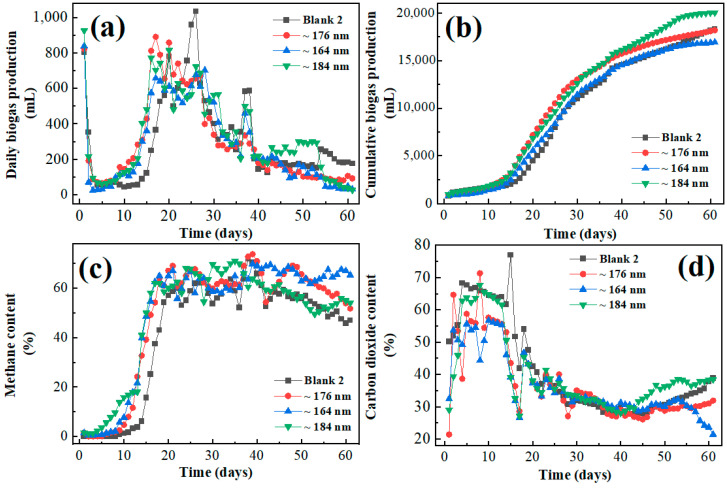
(**a**) Daily biogas production of insect frass with corn straw; (**b**) Cumulative biogas production of insect frass with corn straw; (**c**) Daily methane content of frass with corn straw; (**d**) Daily carbon dioxide content of frass with corn straw.

**Figure 4 biology-13-00536-f004:**
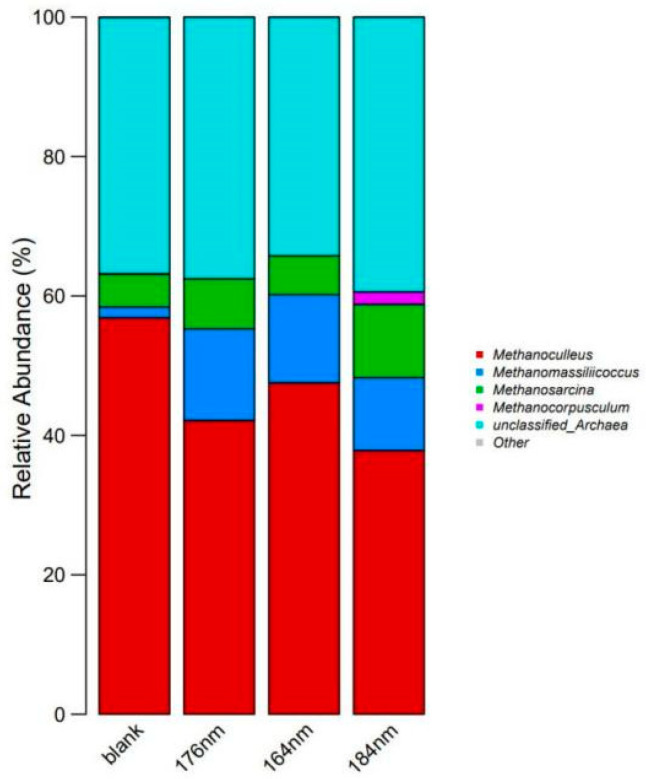
Bar graph of the relative abundance of species in pure frass samples.

**Figure 5 biology-13-00536-f005:**
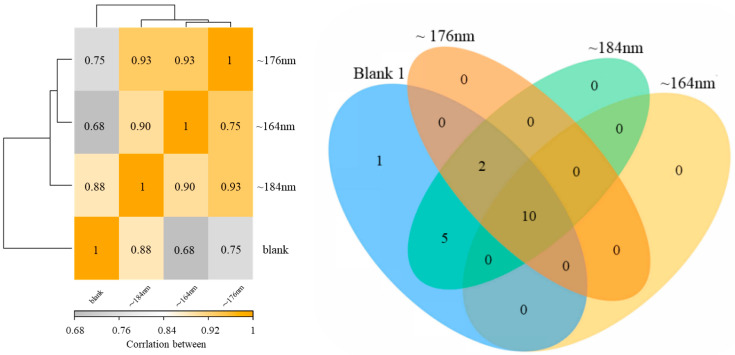
Sample correlation heatmap (the color block represents the correlation index value, and the grayer the color, the more it represents; the lower the correlation index between samples, the yellow the color, and the higher the correlation index); Venn diagram of species distribution represented by different colors, the numbers in the figure represent the number of specific or shared species.

**Table 1 biology-13-00536-t001:** Table of species abundance of each sample.

Species	Blank 1 (/%)	~176 nm (/%)	~164 nm (/%)	~184 nm (/%)
*Methanoculleus*	56.88	42.15	47.54	37.83
*Methanomassiliicoccus*	1.53	13.13	12.64	10.46
*Methanosarcina*	4.70	7.19	5.58	10.51
*Methanocorpusculum*	0.08	0	0	1.78
Unclassified Archaea	36.76	37.53	34.22	39.42
Other	0.04	0	0.01	0.01

**Table 2 biology-13-00536-t002:** Alpha diversity index statistics.

Sample	Number	OTUs	Shannon	Chao	Ace	Simpson	Shannon Even	Coverage
Blank 1	21,181	18	1.36	18	18	0.37	0.47	1
~176 nm	19,980	10	1.61	10	0	0.25	0.70	1
~164 nm	16,690	12	1.60	12	12	0.27	0.64	1
~184 nm	25,551	17	1.81	17	17.75	0.22	0.64	1

Shannon: To estimate the microbial diversity index in a sample. The larger the Shannon value, the higher the community diversity. Chao: To estimate the index of the number of OTUs in a community, which is commonly used in ecology to estimate the total number of species.

**Table 3 biology-13-00536-t003:** FAPROTAX function table for predicting microbial communities in the database.

Group	Blank 1	~176 nm	~164 nm	~184 nm
Methanotrophy	0	0	0	0
Acetoclastic methanogenesis	0	0	0	0
Methanogenesis by disproportionation of methyl groups	995	1436	931	2685
Methanogenesis using formate	0	0	0	0
Methanogenesis by CO_2_ reduction with H_2_	13,061	9858	8866	12,805
Methanogenesis by reduction of methyl compounds with H_2_	325	2623	2110	2673
Hydrogenotrophic methanogenesis	13,386	12,481	10,976	15,478
Methanogenesis	13,386	12,481	10,976	15,478
Methanol oxidation	0	0	0	0
Methylotrophy	325	2623	2110	2673
Aerobic ammonia oxidation	0	0	0	0
Aerobic nitrite oxidation	0	0	0	0
Nitrification	0	0	0	0
Sulfate respiration	0	0	0	0

## Data Availability

Data are contained within the article and Appendix A.

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
