# Peer review of "Impact of Iron Oxide on Anaerobic Digestion of Frass in Biogas and Methanogenic Archaeal Communities’ Analysis"

_biology, 2024, doi:10.3390/biology13070536_

Round 1

Reviewer 1 Report

Comments and Suggestions for Authors

Please take note of the following suggestions for improving the manuscript:

1. Review and condense the text to highlight the important points in the results, making it easier for the reader to follow.

2. Clearly articulate what sets this study apart from similar research. Emphasize any significant novel findings to the reader.

3. Enhance the overall academic quality of the text by avoiding non-academic phrasing and terminology such as "In a nutshell" and "In the realm."

4. Explain the effects of adding different iron oxides on the concentrations of heavy metals or trace elements in various batch tests. Address the potential inhibitory effects resulting from high concentrations of these toxic elements on microbial communities. These issues should be clarified in the updated version of the manuscript.

Comments on the Quality of English Language

1. Review and condense the text to highlight the important points in the results, making it easier for the reader to follow.

2. Clearly articulate what sets this study apart from similar research. Emphasize any significant novel findings to the reader.

3. Enhance the overall academic quality of the text by avoiding non-academic phrasing and terminology such as "In a nutshell" and "In the realm."

Author Response

Comments 1: Review and condense the text to highlight the important points in the results, making it easier for the reader to follow.
Response 1: The language has been further refined and edited.

Comments 2: Clearly articulate what sets this study apart from similar research. Emphasize any significant novel findings to the reader.
Response 2: The language has been further refined and edited.

Comments 3:  Enhance the overall academic quality of the text by avoiding non-academic phrasing and terminology such as "In a nutshell" and "In the realm."
Response 3: The aforementioned sections have been refined and edited for linguistic precision.

Comments 4: Explain the effects of adding different iron oxides on the concentrations of heavy metals or trace elements in various batch tests. Address the potential inhibitory effects resulting from high concentrations of these toxic elements on microbial communities. These issues should be clarified in the updated version of the manuscript.
Response 4: In the experiment, three different sizes of iron oxide nanoparticles were investigated, each with an average size difference of less than 20 nm. These nanoparticles exhibited varying effects on promotion and inhibition. The smallest, at 164 nm, inhibited both methane and carbon dioxide production. The 176 nm nanoparticles promoted methane production while inhibiting carbon dioxide production. The 184 nm nanoparticles enhanced both biogas yield and methane content, with a slight promotion of CO2 production.

Reviewer 2 Report

Comments and Suggestions for Authors

Article Title: Impact of iron oxide on anaerobic digestion of frass in biogas and methanogenicarchaeal communities’ analysis

The manuscript presented requires minor revision for the manuscript to be accepted.

·         The characteristics of both the feedstocks are not presented in the manuscript

·         The C/N ratio of the frass is too low. How is the C/N ratio maintained in the digester

·         3.3 recheck the title of the section

·         The biogas production from other reactors is low when compared to the blank reactor. Put forward a valid reason as minute addition of Fe3O4 will have a minimal effect on biogas production towards the increasing side

·         The authors didn’t mention the VS reduction of each reactor

·         The iron oxide nanoparticles promote the syntrophy between the Bacteria and Methanogenicarchaea. The authors opted out the bacterial population.

·         How do you support your data regarding the syntrophy between the bacteria and archaea?

·         Table 3. The authors shows the acetoclastic methanogenesis as “0” which is not possible cause minimal accetoclasticmethanogensis occurs in each reactor. How does the authors defend the above statement?

·         All the microorganisms names should be in italics as per the nomenclature guidelines.

·         Representation of the impact of iron oxide nanoparticles in a biochemical pathway can be appreciated

·         More information about how these nanoparticles act on the enzyme co-enzyme complexes to enhance the metabolic activity can give additional strength the manuscript

·         Very less information about the nanoparticles and their role on the microbial organisms is mentioned in the manuscript. Please provide in detailed information along with proper references which correlate the information provided in the manuscript

Comments on the Quality of English Language

The language presented in the manuscript is up to the mark and a little modification can be made according to the context. Rest is fine

Author Response

Comments 1: The manuscript presented requires minor revision for the manuscript to be accepted.
Response 1: The language has been further refined and edited.

Comments 2:  The characteristics of both the feedstocks are not presented in the manuscript.
Response 2: The properties of the two materials have been detailed in the Supporting Information tables.

Comments 3:   The C/N ratio of the frass is too low. How is the C/N ratio maintained in the digester
Response 3: This study represents the first investigation into the fermentation potential of pig manure vermicompost. Due to its low carbon-to-nitrogen ratio, it demonstrated relatively satisfactory biogas production. To enhance the carbon content, straw with a high carbon source was added, adjusting the mixture to an optimal C/N. However, the dry straw, which was not pretreated and contained high lignin content, proved resistant to microbial degradation, resulting in a limited contribution to biogas production. Future studies will consider the application of cellulase to degrade the straw’s lignin structure.

Comments 4: recheck the title of the section
Response 4: The section titles have been supplemented and revised accordingly.

Comments 5:  The biogas production from other reactors is low when compared to the blank reactor. Put forward a valid reason as minute addition of Fe3O4 will have a minimal effect on biogas production towards the increasing side
Response 5: Previous literature reports have indicated that methanogenic archaea exhibit a high sensitivity to Fe3O4 nanoparticles, facilitating enhanced direct interspecies electron transfer (DIET) and subsequently promoting methane production. This phenomenon was corroborated in our experiment with the 176 nm and 184 nm nanoparticle groups.

Comments 6:  The authors didn’t mention the VS reduction of each reactor
Response 6: The reduction in VS within the reactor has been supplemented with additional data.

Comments 7: The iron oxide nanoparticles promote the syntrophy between the Bacteria and Methanogenicarchaea. The authors opted out the bacterial population.
Response 7: Nanoparticles of iron oxide have been shown to enhance direct interspecies electron transfer (DIET) among microorganisms. An analysis of the microbial interactions has been included.

Comments 8:  How do you support your data regarding the syntrophy between the bacteria and archaea?
Response 8: The experimental data supporting syntrophic interactions between bacteria and archaea require sophisticated instrumentation, which is currently beyond the capabilities of our laboratory.

Comments 9:   Table 3. The authors shows the acetoclastic methanogenesis as “0” which is not possible cause minimal accetoclasticmethanogensis occurs in each reactor. How does the authors defend the above statement?
Response 9: This data was returned by the sequencing company and has not been modified. They indicated that the data is not entirely accurate and is merely predictive. For precise predictions, metagenomic analysis is required. Given the presence of the acetoclastic methanogenesis pathway, should we delete the rows where the data shows zero?

Comments 10:  All the microorganisms names should be in italics as per the nomenclature guidelines.
Response 10: The names of microorganisms in the text have been revised to be italicized.

Comments 11:  Representation of the impact of iron oxide nanoparticles in a biochemical pathway can be appreciated
Response 11: Additional explanations have been incorporated into the text.

Comments 12: More information about how these nanoparticles act on the enzyme co-enzyme complexes to enhance the metabolic activity can give additional strength the manuscript
Response 12: Additional explanations regarding the hypothesis have been incorporated into the text.

Comments 13: Very less information about the nanoparticles and their role on the microbial organisms is mentioned in the manuscript. Please provide in detailed information along with proper references which correlate the information provided in the manuscript
Response 13:Additional explanations have been incorporated into the text.